# Factors associated with modern contraceptive use among men in Pakistan: Evidence from Pakistan demographic and health survey 2017-18

Ahmad Ali [1]*, Abu Zar[1], Ayesha Wadood[2]

1 Department of Medicine, Nishtar Medical University, Multan, Pakistan, 2 Nuclear Institute for Agriculture and Biology, Faisalabad, Pakistan

☯ These authors contributed equally to this work.

* ahmadumair188@gmail.com

**Data Availability Statement:** All relevant data are within the paper and its Supporting Information files.

## Abstract

### Objective

The role of men in family planning is critical in patriarchal societies like Pakistan. The objective of this study is to explore the predictors of modern contraceptive use among Pakistani men.

### Methods

This study is a secondary analysis of Pakistan demographic and health survey (PDHS) 2017–18 data. The study sample consists of 3691 ever married men aged 15–49 years. Pearson's chi square test and logistic regression were used to find out the determinants of modern contraceptive use among men. Data analysis was carried out in December, 2020.

### Results

Findings of logistic regression showed that men who were uneducated (aOR = 0.746; 95% CI = 0.568–0.980), residing in Sindh (aOR = 0.748; 95% CI = 0.568–0.985), Baluchistan (aOR = 0.421; 95% CI = 0.280–0.632) or FATA (aOR 0.313; 95% CI 0.176–0.556) and those who belonged to the poorest wealth quintile (aOR = 0.569; 95% CI = 0.382–0.846) were less likely to use modern contraceptives. Men who did not wish for another child (aOR = 2.821; 95% CI = 2.305–3.451) had a higher likelihood of modern contraceptive use. Finally, men who thought that contraception was women's business (aOR = 0.670; 95% CI = 0.526–0.853) and those who did not discuss family planning with health worker (aOR = 0.715; 95% CI = 0.559–0.914) were also less likely to use modern contraceptives.

### Conclusion

Reproductive health education of males, targeting males, in addition to, females for addressing family planning issues and improvement of family planning facilities in

**Funding:** The authors received no specific funding for this work.

**Competing interests:** The authors have declared that no competing interests exist.

**Abbreviations:** OR, = odds ratio; 95% CI, 95% confidence interval; aOR, adjusted odds ratio; KPK, Khyber Pakhtunkhwa; FATA, Federally administered tribal areas.

socioeconomically under-privileged regions are suggested to improve contraceptive use among couples.

## Introduction

Contraception plays a pivotal role in the improvement of sexual and reproductive health. It also serves to reduce maternal deaths resulting from unsafe abortions and unwanted pregnancies [1]. A significant association has been observed between fertility reduction and decrease in infant, child and maternal mortality [2]. It was observed that about 13% of infant deaths, 25% of under-five mortalities, and 35% of maternal deaths could be prevented by increasing birth interval by three years [3].

Comparing the Data from Pakistan demographic and health survey (PDHS) conducted in 2017–18 to that of PDHS 2012–13, it becomes evident that the use of modern contraceptives did not improve during this period (35% in PDHS 2012–13 and 34% in PDHS 2017–18) [4]. The contraceptive prevalence rate in Bangladesh (2014) and India (2015) is 62.4% and 52.4% respectively [5, 6].

Pakistan is situated in south Asia and is among the top ten most populous countries in the world. Population growth rate of Pakistan is 2% [7]. Government of Pakistan has taken several measures to reduce population growth rate. Among these measures, improvement in family planning services is worth mentioning. The annual expenditure of family planning services is about US $55 per woman [8]. Despite these efforts the uptake of contraceptives by couples remains low in Pakistan. The unmet need of family planning reported in PDHS 2017–18 was 17% which is less than that reported in PDHS 2012–13 (20%) [4].

Various factors like socioeconomic status, education, cultural beliefs, area of residence, religion and wrong perceptions about family planning determine the utilization of modern contraception [9]. In agriculture based societies, men usually wish to have large number of children because they serve as a source of livelihood. This perception of men creates hinderance in the utilization of contraceptives by couples [10–12]. Studies have also reported that educated parents can better perceive the benefits of having fewer children. In addition, better contraceptives uptake in urban areas may be accounted for by the availability of better health care services and access to information [13]. A study conducted in Karachi reported various myths and beliefs that may lead to reduced contraceptive uptake by couples. These included, perceived adverse effects of contraceptives like weight gain, birth defects, infertility and reduced sexual pleasure. Furthermore, cultural disapproval and perception that practicing contraception leads to displeasure of God were found to be prevalent [14].Decisions about the number of children are largely dependent on males in patriarchal societies like Pakistan [15, 16]. International conference on population and development held in Cairo (1994) highlighted the importance of involving males in the issues of reproductive and sexual health. Despite this, participation of males in family planning issues remains limited [17, 18].

Pakistan also participated in the international conference on population and development (ICPD) held in Nairobi. During this summit, Pakistan committed to achieve universal sexual and reproductive health coverage, increase contraceptive prevalence rate to 60% by 2030, finance ICPD programs and to eliminate gender based violence. However, Engagement of males in family planning programs was not specifically targeted [19]. Research studies exploring the predictors of contraceptive use among couples have overlooked the role of men [20]. This indirectly endorses the concept that contraception is the responsibility of females [21, 22].

Because of a generalized perception that contraception is women's affair, men have remained excluded from family planning matters [23].

This study attempts to explore the factors determining the use of modern contraceptive methods among Pakistani men using PDHS 2017–18 dataset. The results of this analysis will be significant not only for the Government of Pakistan but also for various non-government organizations that are working to promote family planning. In addition, this paper also puts emphasis on the fact that the objectives of family planning programs may not be accomplished without considering the role of men as decision makers in the matters of family planning and reproductive health in patriarchal societies.

## Methods

### Data source

This study is a secondary analysis of Pakistan Demographic and Health Survey (PDHS) 2017–18. PDHS is a cross sectional survey that reports health and demographic features of Pakistani population. The survey was conducted by national institute of population studies (NIPS), Pakistan. Data collection during the survey was done as follows. Eight regions viz Punjab, Sindh, Khyber Pakhtunkhwa, Baluchistan, Azad Jammu and Kashmir, Gilgit Baltistan, Islamabad and Federally Administered Tribal Areas (FATA) were surveyed. Each region was divided into urban and rural areas creating sixteen strata. This was followed by the selection of clusters from each stratum using probability sampling. A total of 580 clusters were selected. Nineteen of them were abandoned due to security risk. The final number of clusters became 561. Selection of clusters was followed by the sampling of households from each cluster. Systematic random sampling was used for this purpose. A fixed number of twenty-eight households were selected from each cluster. Eventually, 16,240 households were surveyed.

This survey used six questionnaires namely household questionnaire, women's questionnaire, men's questionnaire, fieldworker's questionnaire, biomarker questionnaire and community questionnaire. Men's questionnaire was used to interview ever married men aged 15–49 years. A total of 3691 ever married men aged 15–49 years across Pakistan (including Azad Jammu Kashmir and Gilgit Baltistan) were interviewed in this survey. This secondary analysis was carried out using the data of 3691 ever married men aged 15–49 years in December, 2020.

Ethical approval for PDHS 2017–18 was obtained from the National Bioethics Committee, Pakistan Health Research Council and the International Review Board of ICF. For more details on PDHS sample collection procedure, please refer to the PDHS 2017–18 report [4]. Owing to secondary nature of this analysis, no informed consent or ethical approval was needed. However, permission to use the data for this study was acquired from ICF International after providing a brief description of the study.

**Dependent variable.** The outcome variable is "modern contraceptive usage". It is a binary variable with two categories encoded as '0' (men who reported no modern contraceptive use) and '1' (men who reported the use of modern contraceptive methods by themselves or their female partners). Contraceptive methods categorized as 'modern' in PDHS include pills, injections, intrauterine devices, implants, male/female condoms, diaphragms, foam, jellies, female/male sterilization, emergency contraception and lactational amenorrhea. Contraceptive methods categorized as 'traditional' include withdrawal, periodic abstinence and abstinence. The third category of contraceptive methods described in PDHS is 'folkloric methods' which are country specific.

**Independent variables.** Explanatory variables selected for this study belonged to four different categories including sociodemographic factors, socioeconomic factors, behavioral factors and communication with health system. Fig 1 demonstrates a conceptual framework of

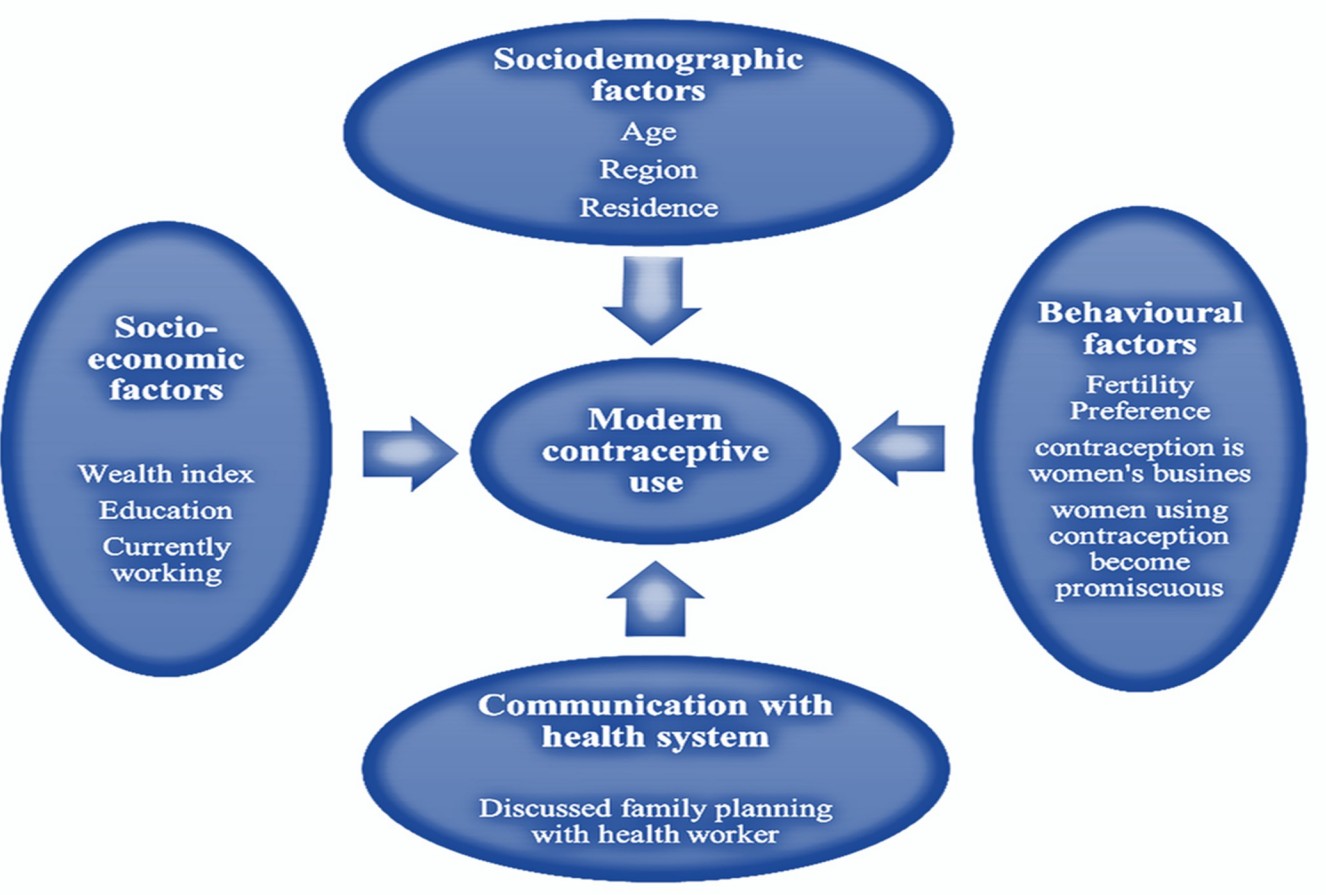

**Fig 1. Schematic representation of study variables.**

the variables used in this study. Demographic factors were Age, region (Punjab, Sindh, Khyber Pakhtunkhwa, Baluchistan, Islamabad, Gilgit Baltistan, Azad Jammu Kashmir and FATA) and residence (urban/rural). Socioeconomic variables selected for the study were education, household wealth index and employment status (obtained from the question if the respondent was currently working). Behavioral factors included fertility preference (wish for additional children/ no wish for more children/ sterilized or partner-less) and belief in certain false notions like "Contraception is women's business" and "women who use contraception become promiscuous." Communication with health system was assessed by asking the respondents if they discussed family planning with a health worker.

The variables "age" and "education were recoded. Age was divided into three categories; 15–29 years, 30–45 years and 45 or above. Education level was divided into three categories: no education, primary education, secondary education or higher. Wealth index (poorest/ poorer/middle/richer/richest) was calculated during the survey by using data on household assets (e.g. type of vehicle owned, possession of various electronic appliances et c.) and other household characteristics (e.g. building material of the house etc.).

## Statistical analysis

Data analysis was performed using IBM SPSS version 21. The data was weighted using men's sample weights in PDHS dataset. Descriptive statistics were applied to obtain a summary of

background characteristics of the respondents. Initially, Pearson's chi squared test was used to determine the relationship between modern contraceptive use and each predictor variable. All the independent variables, except employment status (Currently working or not), were significantly associated with the outcome and were selected for multivariable logistic regression. Before running logistic regression, all the covariates were analyzed for the presence of multicollinearity using tolerance statistics and variance inflation factor "S1 Table". No evidence of multicollinearity was detected. Bivariable logistic regression was run first. Finally, multivariate binary logistic regression was used by adjusting for the covariates to determine the impact of predictor variables on modern contraceptives use among Pakistani men. A p value of less than 0.05 was considered statistically significant.

The choice of reference category in logistic regression was guided by relevant literature. In addition, the reference categories were selected based on the hypothesis e.g. "residents of Punjab" and "educated men" were anticipated to have a higher modern contraception uptake. Thus, these were selected as reference categories to check how these categories differ relative to their counterparts.

## Results

### Background characteristics of the respondents

The findings of descriptive statistics are presented in Table 1. The respondents included in the analysis were 3691 ever married men age 15–49 years. Only about 19.3% individuals reported that they were using a modern contraceptive method. Nearly half (49%) of men were residing in rural areas. About a quarter (23.1%) of men belonged to the province Punjab while the proportion of respondents from Sindh, KPK and Baluchistan was 21.1%, 13.7% and 14.1% respectively. The level of education of more than half (59.4%) of men was secondary or higher. Less than one fourth (23.5%) of the respondents received no education. Each wealth quintile had an approximately similar number of respondents. Only 5.2% of the men had no employment (not currently working). More than half (59.5%) of the individuals responded that they wanted to have another child. About one third (33.7%) of the men said that they no more wished for another child. Only 12.4% men discussed family planning with a health care worker within a few months preceding the survey. About one quarter of men (25.1%) agreed to the notion that contraception was women's business. About 1 in every ten men (9.9%) believed that using contraception could make women promiscuous.

Table 2 presents the findings of Pearson's Chi square test. Age, region, residence, education, wealth index, fertility preference, discussion with health worker and the behavioral factors (belief in the notions "contraception is women's business" and "women using contraception become promiscuous") were significantly associated with modern contraceptive usage (p<0.001). The variable 'currently working' did not have a significant relation with the usage of modern contraceptive methods.

### Determinants of modern contraceptive use

Table 3 presents the findings of logistic regression. The odds of modern contraception usage was significantly low in the residents of Sindh (aOR = 0.748; 95%; CI = 0.568–0.985), Baluchistan (aOR = 0.421; 95% CI = 0.280–0.632) and FATA (aOR = 0.313; 95% CI = 0.176–0.556) as compared to those of Punjab. The respondents who received no education had lesser odds (aOR = 0.746; 95% CI = 0.568–0.980) of using modern contraceptive methods when compared to those with primary, secondary or higher level of education. The odds of modern contraceptive usage in men belonging to the poorest wealth quintile (aOR = 0.569; 95% CI = 0.382–0.846) were significantly less relative to other wealth quintiles. Higher odds of modern

**Table 1. Demographic, socioeconomic and behavioral characteristics of men.**

| Variables | | Percentage | Frequency |
|---|---|---|---|
| Age | 15–29 | 27.5 | 1014 |
| | 30–44 | 56.4 | 2081 |
| | 45 and above | 16.1 | 596 |
| Residence | Urban | 51.0 | 1884 |
| | Rural | 49.0 | 1807 |
| Region | Punjab | 23.1 | 853 |
| | Sindh | 21.1 | 778 |
| | KPK | 13.7 | 505 |
| | Baluchistan | 14.1 | 522 |
| | Islamabad | 7.2 | 265 |
| | Gilgit Baltistan | 5.7 | 210 |
| | Azad Jammu Kashmir | 9.1 | 336 |
| | FATA | 6.0 | 222 |
| Education | None | 23.5 | 869 |
| | Primary | 17.0 | 628 |
| | Secondary or higher | 59.4 | 2194 |
| Wealth index | Poorest | 18.2 | 672 |
| | Poorer | 21.7 | 801 |
| | Middle | 19.2 | 708 |
| | Richer | 19.6 | 725 |
| | Richest | 21.3 | 785 |
| Currently working | Yes | 94.8 | 3496 |
| | No | 5.2 | 191 |
| Fertility preference | Have another | 59.5 | 2154 |
| | No more | 33.7 | 1221 |
| | Undecided | 4.7 | 168 |
| | Sterilized/infertile/no partner | 2.1 | 76 |
| Discussed family planning with health worker | Yes | 87.6 | 459 |
| | No | 12.4 | 3231 |
| Contraception is woman's business | Agree | 25.1 | 928 |
| | Disagree | 65.7 | 2426 |
| | Don't know | 9.1 | 336 |
| Women who use contraception become promiscuous | Agree | 9.9 | 365 |
| | Disagree | 76.4 | 2818 |
| | Don't know | 13.7 | 506 |
| Modern contraception usage | Yes | 19.3 | 714 |
| | No | 80.6 | 2977 |
| Total | | 100.0 | 3691 |

contraceptive usage were found in those men who showed no desire to have another child (aOR = 2.821; 95% CI = 2.305–3.451). Those respondents who had no discussion related to family planning with a health worker were significantly less likely (aOR = 0.715; 95% CI = 0.559–0.914) to use modern contraceptive methods. Finally, the men who agreed (aOR = 0.670; 95% CI = 0.526–0.853) to the concept that contraception was women's business had significantly lesser odds of using modern contraceptive methods. In addition, the men who were reportedly sterilized, infertile or currently partner-less had higher odds

**Table 2. Association of independent variables with modern contraceptive use.**

| Variables | | Modern contraceptive use (%) | | P value |
|---|---|---|---|---|
| | | No | Yes | |
| Age | 15–29 | 87.2 | 12.8 | **0.001** |
| | 30–44 | 78.4 | 21.6 | |
| | 45 and above | 75.4 | 24.6 | |
| Residence | Urban | 77.5 | 22.5 | **0.001** |
| | Rural | 83.3 | 16.7 | |
| Region | Punjab | 75.8 | 24.2 | **0.001** |
| | Sindh | 83.5 | 16.5 | |
| | KPK | 75.4 | 24.6 | |
| | Baluchistan | 92.6 | 7.4 | |
| | Islamabad | 70.5 | 29.5 | |
| | Gilgit Baltistan | 73.3 | 26.7 | |
| | Azad Jammu Kashmir | 75.8 | 24.2 | |
| | FATA | 93.2 | 6.8 | |
| Education | None | 87.8 | 12.2 | **0.001** |
| | Primary | 79.1 | 20.9 | |
| | Secondary or higher | 77.7 | 22.3 | |
| Wealth index | Poorest | 90.6 | 9.4 | **0.001** |
| | Poorer | 83.5 | 16.5 | |
| | Middle | 77.8 | 22.2 | |
| | Richer | 75.8 | 24.2 | |
| | Richest | 74.6 | 25.4 | |
| Currently working | Yes | 80.1 | 19.9 | 0.255 |
| | No | 83.7 | 16.3 | |
| Fertility preference | Have another | 87.4 | 12.6 | **0.001** |
| | No more | 69.8 | 30.2 | |
| | Undecided | 84.0 | 16.0 | |
| | Sterilized/infertile/no partner | 39.5 | 60.5 | |
| Discussed family planning with health worker | Yes | 72.4 | 27.6 | **0.001** |
| | No | 81.5 | 18.5 | |
| Contraception is woman's business | Agree | 85.4 | 14.6 | **0.001** |
| | Disagree | 76.8 | 23.2 | |
| | Don't know | 91.8 | 8.2 | |
| Women who use contraception become promiscuous | Agree | 80.8 | 19.2 | **0.001** |
| | Disagree | 79.0 | 21.0 | |
| | Don't know | 87.0 | 13.0 | |

(aOR = 11.224; 95% CI = 6.706–18.784) of modern contraceptive usage compared to those who wanted more children.

## Discussion

The policy paper of Ministry of planning, development and special initiatives 2020 stated various measures to improve contraceptive prevalence rate including development of skilled human resource, adequate supply of modern contraceptives, targeting female population with high risk fertility etc. [24]. Evidence suggests that encouraging men to be supportive partners

**Table 3. Predictors of Modern contraceptive use among men.**

| Variables | | Modern contraceptive use | | | | | |
|---|---|---|---|---|---|---|---|
| | | Bivariate logistic regression | | | Multivariate logistic regression | | |
| | | OR | 95% CI | P value | aOR | 95% CI | P value |
| Age | 15–29 | 1.00 | | | 1.00 | | |
| | 30–44 | 1.89 | 1.52–2.34 | **0.001** | 1.22 | 0.96–1.55 | 0.091 |
| | 45 and above | 2.23 | 1.71–2.91 | **0.001** | 1.00 | 0.73–1.36 | 0.998 |
| Residence | Urban | 1.00 | | | 1.00 | | |
| | Rural | 0.69 | 0.58–0.81 | **0.001** | 0.87 | 0.71–1.07 | 0.202 |
| Region | Punjab | 1.00 | | | 1.00 | | |
| | Sindh | 0.61 | 0.48–0.78 | **0.001** | 0.74 | 0.56–0.98 | **0.039** |
| | KPK | 1.01 | 0.78–1.31 | 0.900 | 1.26 | 0.95–1.68 | 0.101 |
| | Baluchistan | 0.24 | 0.17–0.36 | **0.001** | 0.42 | 0.28–0.63 | **0.001** |
| | Islamabad | 1.30 | 0.95–1.79 | 0.092 | 1.19 | 0.84–1.68 | 0.323 |
| | Gilgit Baltistan | 1.13 | 0.80–1.60 | 0.468 | 1.44 | 0.98–2.13 | 0.060 |
| | Azad Jammu Kashmir | 0.99 | 0.73–1.34 | 0.975 | 1.01 | 0.73–1.40 | 0.933 |
| | FATA | 0.22 | 0.13–0.39 | **0.001** | 0.31 | 0.17–0.55 | **0.001** |
| Education | None | 0.48 | 0.38–0.81 | **0.001** | 0.74 | 0.56–0.98 | **0.035** |
| | Primary | 0.92 | 0.74–1.15 | 0.486 | 1.10 | 0.86–1.41 | 0.430 |
| | Secondary or higher | 1.00 | | | 1.00 | | |
| Wealth quintile | Poorest | 0.30 | 0.22–0.41 | **0.001** | 0.56 | 0.38–0.84 | **0.005** |
| | Poorer | 0.58 | 0.45–0.74 | **0.001** | 0.88 | 0.94–1.20 | 0.421 |
| | Middle | 0.84 | 0.66–1.06 | 0.157 | 1.06 | 0.80–1.39 | 0.681 |
| | Richer | 0.93 | 0.74–1.18 | 0.593 | 1.09 | 0.85–1.41 | 0.466 |
| | Richest | 1.00 | | | 1.00 | | |
| Fertility preference | Have another | 1.00 | | | 1.00 | | |
| | No more | 0.12 | 0.06–0.23 | **0.001** | 2.82 | 2.30–3.45 | **0.001** |
| | Undecided | 0.09 | 0.05–0.15 | **0.001** | 1.42 | 0.90–2.25 | 0.125 |
| | Sterilized/infertile/no partner | 0.28 | 0.17–0.45 | **0.001** | 11.2 | 6.70–18.78 | **0.001** |
| Discussion with health worker about contraception | Yes | 1.00 | | | 1.00 | | |
| | No | 0.59 | 0.47–0.74 | **0.001** | 0.71 | 0.55–0.91 | **0.008** |
| Contraception is women's business | Disagree | 1.00 | | | 1.00 | | |
| | Agree | 0.56 | 0.46–0.69 | **0.001** | 0.67 | 0.52–0.85 | **0.001** |
| | Don't know | 0.29 | 0.19–0.44 | **0.001** | 0.52 | 0.31–0.87 | **0.014** |
| Women using contraception become promiscuous | Disagree | 1.00 | | | 1.00 | | |
| | Agree | 1.78 | 1.34–2.35 | **0.001** | 1.07 | 0.77–1.47 | 0.675 |
| | Don't know | 1.59 | 1.09–2.30 | **0.014** | 1.05 | 0.73–1.51 | 0.778 |

for contraceptives use improves health outcomes. More recently, the role of men in family planning is being evolved from "supportive partners" to active users of family planning services and to improve their own reproductive health as well. This indicates why there is a pressing need to approach men in family planning and reproductive health matters [25].

This study examined the factors associated with modern contraceptive use among Pakistani men using the data of Pakistan demographic and health survey 2017–18. Education, region, socioeconomic status, fertility preference, the perception that contraception is women's business and discussion of family planning with health workers were significant predictors of modern contraceptive usage in Pakistani men.

Findings of this paper highlighted that men who discussed family planning with a healthcare worker had a higher likelihood of using modern contraceptive methods. This finding is in

line with the findings from a study conducted in Uganda. Evidence from behavior change models suggests that knowledge is the first step towards change in behavior [26]. Discussion with a health worker about family planning enhances the knowledge of contraception that ultimately brings a positive change in behavior [27]. Similar findings were also reported by studies conducted in Congo [28] and Tanzania [29].

Wealth index also came out to be a significant predictor of modern contraceptive use. Findings from a study conducted in Myanmar also suggested a lower likelihood of modern contraceptive use among men belonging to poor wealth quintile [30]. In addition, the findings of DHS surveys conducted in 18 countries of Latin America, Africa, Caribbean and Asia also supported this finding [31].

In agreement with the findings of this paper, a study conducted in Uganda found that men with higher education were more likely to use modern contraception compared to those who received no formal education [32]. Education helps to get employment which, in turn, increases "household income": a predictor of modern contraceptive use [32]. Certain other studies also reported similar findings on education and wealth index as predictors of modern contraceptive use [33, 34].

This study found significantly less likelihood of modern contraceptive use among men residing in Baluchistan and FATA. A study from Pakistan showed lesser use of contraceptive methods among the residents of Baluchistan relative to other provinces especially Punjab. The probable explanation for this finding is improved structure of family planning services in Punjab [13]. A report was published on the ranking of various provinces of Pakistan based on conduciveness to family planning services. The report discussed various socio-demographic parameters that are linked to contraception uptake, e.g. a better female literacy rate in Punjab (62%) compared to other provinces of Pakistan (Sindh 44%, KPK 35% and Baluchistan 16%). In addition, better utilization of antenatal care services and a higher proportion of economically active women, as described in the said report, may also partly explain why Punjab outperformed in contraceptives uptake relative to other regions [35]. Another previous study also reported that the use of modern contraception varied across various regions of the country [36].

Men who showed no desire for another child were more likely to report modern contraceptive use in a study carried out in Kenya, a finding consistent with the results of this study [37]. Another study showed that men who were infertile or had no partner were more likely to use modern contraceptive methods when compared to those who wanted additional children [1]. Men who do not have a legit partner are more likely to use contraception in order to avoid fathering an illegitimate child which is socially unacceptable [38].

This study also found that men who believed that contraception was women's business were significantly less likely to use modern contraceptive methods. This finding is in compliance with a study from Nigeria which states that negative attitudes of men towards family planning adversely affect modern contraceptive uptake among women [39].

The findings of this study are influenced by recall bias, which represents a potential limitation of the study. In addition, a causal relationship between outcome and predictors cannot be established due to cross-sectional nature of this survey. There were various potential independent variables like parity, sexual activity and marital status which could be incorporated in this analysis. However, the data on these variables could not be retrieved from the dataset. Moreover, This study measured how various predictors influence men's behavior to use contraception. It did not differentiate between men who used male methods from those who relied solely on female methods of contraception. This represents a potential area which can be explored in future studies.

## Conclusion

The role of men in family planning is critical in patriarchal societies like Pakistan. Men who are uneducated, belong to poor socioeconomic status, residents of underprivileged areas, and men having false perceptions (contraception is women's business) are less likely to use contraception. Educating males about reproductive health to eradicate false perceptions about contraception are recommended to improve the efficiency of family planning services in Pakistan. In addition, women empowerment and reproductive health education will also improve contraceptive uptake by couples. Appropriate policy needs to be formulated to ensure the engagement of men in family planning programs, e.g. by encouraging the involvement of men in family planning discussions or counselling sessions. This is because the objectives of family planning programs may not be met without considering the role of men as chief decision makers in patriarchal societies. The findings of this study may contribute to the achievement of Pakistan's commitment to increase the contraceptive prevalence rate by 60% in 2030.

## Supporting information

**S1 Checklist. STROBE checklist.**
(DOC)

**S1 Table. Findings of collinearity diagnostics (tolerance statistics and variance inflation factor).**
(DOCX)

**S2 Table. Descriptive table on various contraceptive methods.**
(DOCX)

**S1 File.**
(SAV)

## Acknowledgments

Authors are grateful to Measures DHS for granting access to use PDHS 2017–18 dataset for this research.

## Author Contributions

**Conceptualization:** Ahmad Ali, Abu Zar.

**Data curation:** Ahmad Ali, Abu Zar.

**Formal analysis:** Ahmad Ali, Abu Zar, Ayesha Wadood.

**Methodology:** Ahmad Ali, Ayesha Wadood.

**Validation:** Ayesha Wadood.

**Writing – original draft:** Ahmad Ali, Abu Zar, Ayesha Wadood.

**Writing – review & editing:** Ayesha Wadood.

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
