## [Decision Letter · Decision Letter 0]

29 Mar 2022

PONE-D-21-23985

Determinants of modern contraceptive use among men in Pakistan: evidence from Pakistan demographic and health survey 2017-18

PLOS ONE

Dear Dr. Ali,

Thank you for submitting your manuscript to PLOS ONE. After careful consideration, we feel that it has merit but does not fully meet PLOS ONE’s publication criteria as it currently stands. Therefore, we invite you to submit a revised version of the manuscript that addresses the points raised during the review process.

Three reviewers have provided detailed comments on the manuscript that requires changes including in the language. For completeness of reporting, PLOS ONE strongly suggests the inclusion of a relevant checklist for observational study such as the STROBE checklist (https://www.strobe-statement.org).

We look forward to receiving your revised manuscript.

Kind regards,

José Antonio Ortega, Ph.D.

Academic Editor

PLOS ONE

A clean copy of the edited manuscript (uploaded as the new *manuscript* file).

Reviewers' comments:

Reviewer's Responses to Questions

**Comments to the Author**

1. Is the manuscript technically sound, and do the data support the conclusions?

Reviewer #1: Yes

Reviewer #2: Yes

Reviewer #3: Partly

2. Has the statistical analysis been performed appropriately and rigorously? 

Reviewer #1: Yes

Reviewer #2: Yes

Reviewer #3: No

3. Have the authors made all data underlying the findings in their manuscript fully available?

Reviewer #1: Yes

Reviewer #2: Yes

Reviewer #3: Yes

4. Is the manuscript presented in an intelligible fashion and written in standard English?

Reviewer #1: Yes

Reviewer #2: Yes

Reviewer #3: No

5. Review Comments to the Author

Reviewer #1: The research article “Determinants of modern contraceptive use among men in Pakistan: evidence from Pakistan demographic and health survey 2017-18” uses secondary data analysis of Pakistan Demographic and Health Survey (DHS) to explore the determinants of modern contraceptive use among men.

The study is important as the unmet need for family planning in Pakistan is high and male´s attitudes regarding contraception are an important part of fertility rates.

Background

Consider changing the word “determinants” to “factors associated with”, as DHS do not have a longitudinal design, thus cannot be utilized to study causality.

The author says that U$ 55 are spent per woman, but it is not clear if it is a yearly rate.

As much as the authors cite the relevant factors that matter for the utilization of modern contraceptive among men and bring some of these factors in the discussion (socioeconomic status, education, cultural beliefs, area of residence, religion and wrong perceptions about family planning), it would be interesting to hear more about these mechanisms up front. I would add a section containing a brief review of these factors, especially because testing these elements is the main goal of these analyses.

The same goes for research studies “exploring the predictors of contraceptive use among couples have overlooked the role of men”. The authors should bring more of this literature and how their study contributes to this framework.

The following phrase is misplaced: “In agriculture based societies, men usually wish to have large number of children because they serve as a source of livelihood. This perception of men creates hinderance in the utilization of contraceptives by couples [18][19][20]”. This is an example of a mechanism and should be included in the section I suggest you add.

The findings would also benefit from more background regarding the regions explored (Punjab, Sindh, KPKa, and so on). Why would I expect regions to differ regarding contraception use? Could levels of development and indicators of gender equity (such as female illiteracy rate) be used to explain some of these differences observed? What has the literature said of these places? You explain Punjab may have improved the structure of family planning services. I would like to hear more about it.

In the end of the Background section, the authors highlight the importance of their findings to frame family planning programs considering the role of men as decision makers in the matters of family planning and reproductive health in patriarchal societies. I think this is gold and should be brought up again in the results with clear and stated recommendations.

I also suggest you enrich your review by bringing information about factors associated with contraceptive use for Pakistan women.

Methods

The analysis was done carefully and the method is adequate for the research question. However, I think it is important to insert controls for parity and if men are sexually active.

I also think it would be good to leave age as a continuous variable, unless you have reasons to believe those three age groups should remain separate.

Being a DHS, it is important to provide information regarding survey weights and sampling.

I understand that calculating a Wealth Index using quintiles would automatically separate the observations into five groups with approximately the same number of respondents. So, these phrase is irrelevant: “Each wealth quintile had an approximately similar number of respondents”.

Regarding the logistic regression, I understood from the text that authors only used in the multivariate analysis the variables that had been found to be significant in the bivariate model. However, by looking at Table 3, all variables were included (indeed, they are all significant at the bivariate model). I would add a phrase explaining that they are all significant. If it is not what you did, please, clarify.

Conclusion

I think this manuscript would benefit from a Review as it has the potential to make a good contribution in its field.

Reviewer #2: The question of fertility plays a vital role in many countries’ economic development and health objectives, including Pakistan. It is, therefore, an important area for investigation and the use of large datasets such as the demographic and health survey (PDHS). I appreciate the authors’ curiosity and diligence in investigating this area. Hopefully, my comments can be used to enhance the work.

Introduction

On page 03, line 51, the reference to fertility reduction as being connected to improvements in health and economic activity is a bit vague. Are you referencing individuals or the state of the country as a whole? Also, it is a bit of a nuanced argument that would require some expansion. However, your study seems more concerned with reaching the male population and less of the significant picture objectives, perhaps focusing on the male perspective. Why have males historically been left out of the conversation on family planning? Why has it taken so long to recognize the importance of reaching out to males in that space? Why have previous efforts tended to focus on women?

On page 4, Line 73 -Line 74, you refer to the United Nations International Conference on Population and Development. That was more than a generation ago, and there have been many iterations of global, regional, and country-specific policies and objectives that have addressed some of the questions you are investigating in this paper. Perhaps you should reference something more recent (e.g., the Millennium Development Goals, followed by the SDGs). Even the UN ICPD had a 25th reunion (i.e., the Nairobi Summit). How have such global structures framed the issue of fertility and contraceptive use behaviors?

Methods

The methods read as straightforward and well-executed. I have nothing to add to this section except for the choice of the conceptual framework. Why was that framework chosen? What is the justification for choosing to look at the variables of interest through that lens? Several conceptual frameworks have been used over the years to explain contraceptive use and family planning behaviors (e.g., USAID’s Conceptual Framework and the Women’s and girls’ empowerment in sexual and reproductive health (WGE-SRH) framework). Why did you eschew other frameworks in favor of this one?

As a note, Figure 01 is blurry, and I think you should find a better image that is clearer and easier to read.

Results

Be careful of the language you employ in describing your results. For example, in the results section, you use the terms likelihood and odds interchangeably. Though they are used to mean the same thing in colloquial usage, “odds” references a particular relationship between the ratio of probabilities (see pg. 10, Line 171 compared to Line 175).

Also, be sure to state that a finding is statistically significant and not just significant. Again, statistically significant as a technical term references something particular, while a finding being “significant” can have a more general meaning, such as being generally important or worth noting.

Discussion

The paper’s discussion section is an excellent place to explore your findings and their potential impact. As I read through this section, it mostly reads as a reiteration of the results section. Try exploring some of the following areas:

Given that in the introduction, you mention that your findings may be used to inform policy, I was hoping the discussion would delve into some commentary about Pakistan’s policy regarding modern contraceptive use if there is one, and how findings can influence that. The fertility challenges have been documented for many decades now and I, as a reader, assume that there are some policies already in place. How do your findings impact those kinds of initiatives? Are men historically excluded from those policies? If men as a subgroup have factored into such initiatives, what do these findings mean?

On pg. 13, Line 217 to Line 222, you note regional differences, particularly between Punjab and other regions. You theorize that it may be due to improve family planning programs. Is it possible to expand a bit on this? What kind of well-structured family planning programs in Punjab make that place more successful than other regions, and why have such policies not been diffused to other regions.

Why do you relate your study findings to findings from places like Uganda and Congo? There is nothing wrong per se with that comparison; I don’t think. You could argue that Pakistan occupies a similar socio-economic bracket as those countries, and hence the way they deal with contraceptive use and family planning behaviors may have some bearing on Pakistan. However, neighboring countries in the subregion (e.g., India, Tajikistan, Afghanistan, and Iran) may offer more forthright comparators. I suggest looking through the literature.

Reviewer #3: Review Comments to the Author

Dear Editor,

Thank you for the opportunity to review the manuscript titled Determinants of modern contraceptive use among men in Pakistan: evidence from Pakistan demographic and health survey 2017-18.

Overall, the manuscript is fairly well written and has clear aims. It also focuses on a topic of deep interest to the reproductive health community and is backed by an extensive body of research. However, I find that the justification for the study and its relative contribution to the existing literature is very weak. As such, the unique contribution of these analysis is unclear to me. I think the authors can make a stronger case if they could point out which variables have not been examined, whether is consensus and variables that have inconsistent results. Please see below my specific comments that could further improve the manuscript.

Introduction

Overall – Fairly clear and well written.

1. There are multiple typos throughout the manuscript. Authors are suggested to proofread them carefully.

2. Consider reporting on how Pakistan compares in relation to regional estimates of modern contraceptive use.

3. Need to define modern contraceptives.

4. If effective strategies need to be country-specific, what was the rationale behind estimating prevalence? The rationale part needs to be strengthened.

Methods

1. Data sources and sampling techniques – could benefit from better organization structure – information seems randomly placed. Also, expand on stratification and provide a reference for a more elaborate description for the DHS sampling strategy.

2. Sample –Provide a justification for your sample selection

3. Please provide another Table and give how each of the independent variables were derived or recoded from the original dataset. This can either be in the manuscript or attached as a supplementary file.

4. Which sampling weight and id were used and was the weight normalized?

5. What informed the inclusion or selection of the independent variables?

6. Please specify the model equation

7. What informed the choice of the reference categories

8. Did the authors check for multi-collinearity, the results should be provided

9. Please use the STROBE guidelines and present it as an appendix or a supplementary file

Results

1. Consider specifying only the key findings of the study rather than listing all the determinants. Also, use the breakdown of the sentences rather than writing one long sentence that is hard to follow.

2. The statistical analysis not been performed rigorously

3. Presentation of results needs to be revised

Discussion

1. Authors started to compare and contrast study findings, which should have been followed after stating the key findings and justification for those findings.

2. There is a repetition of most of the findings that are already stated under the results section.

3. Justification for all the discordant results are presented same i.e., due to differences in sample size, study design, setting, and study population. This needs to be study specific rather than a mere generalization.

Conclusion

1. Clearly and concisely state the conclusions of the study in relation to the key question it sought to answer and the contribution that the paper would make.

2. The conclusion is well presented. However, the policy implications are not well discussed. The authors can consider beefing them up.

6. PLOS authors have the option to publish the peer review history of their article (what does this mean?). If published, this will include your full peer review and any attached files.

Reviewer #1: No

Reviewer #2: No

Reviewer #3: No

---

## [Author Response · Author response to Decision Letter 0]

9 May 2022

Reviewer no. 1

• Consider changing the word “determinants” to “factors associated with”, as DHS do not have a longitudinal design, thus cannot be utilized to study causality.

The suggested change has been made.

• The author says that U$ 55 are spent per woman, but it is not clear if it is a yearly rate.

The suggested change has been made in line no. 61.

• As much as the authors cite the relevant factors that matter for the utilization of modern contraceptive among men and bring some of these factors in the discussion (socioeconomic status, education, cultural beliefs, area of residence, religion and wrong perceptions about family planning), it would be interesting to hear more about these mechanisms up front. I would add a section containing a brief review of these factors, especially because testing these elements is the main goal of these analyses. 

Additional information from relevant literature explaining the mechanisms of various predictors on contraception has been added as per the reviewer’s suggestion (line 73-80).

• The same goes for research studies “exploring the predictors of contraceptive use among couples have overlooked the role of men”. The authors should bring more of this literature and how their study contributes to this framework

Additional information from relevant literature has been added as per the reviewer’s suggestion (line 93-94).

• The following phrase is misplaced: “In agriculture based societies, men usually wish to have large number of children because they serve as a source of livelihood. This perception of men creates hinderance in the utilization of contraceptives by couples [18][19][20]”. This is an example of a mechanism and should be included in the section I suggest you add.

I appreciate reviewer’s efforts and keen interest in my manuscript. The phrase has been repositioned (line no. 71-73).

• The findings would also benefit from more background regarding the regions explored (Punjab, Sindh, KPKa, and so on). Why would I expect regions to differ regarding contraception use? Could levels of development and indicators of gender equity (such as female illiteracy rate) be used to explain some of these differences observed? What has the literature said of these places? You explain Punjab may have improved the structure of family planning services. I would like to hear more about it.

Ministry of national health services regulation and coordination, Pakistan published a report that commented on the ranking of various provinces of Pakistan based on their conduciveness to contraception services. The report pointed out that because of better female literacy rate, women empowerment, better utilization of antenatal care services, contraception prevalence is better in Punjab. This information is incorporated in line no. 261-267. 

• In the end of the Background section, the authors highlight the importance of their findings to frame family planning programs considering the role of men as decision makers in the matters of family planning and reproductive health in patriarchal societies. I think this is gold and should be brought up again in the results with clear and stated recommendations.

I am thankful to the reviewer for highlighting an important area of the introduction and its relevance to results and conclusion.

• I also suggest you enrich your review by bringing information about factors associated with contraceptive use for Pakistan women

The study intends to look for the factors associated with modern contraceptive use in men. Authors have tried to put together the most pertinent information from the literature. Additional information on women related factors may unduly prolong the introduction section and may appear disconnected from the rest of the literature.

• The analysis was done carefully and the method is adequate for the research question. However, I think it is important to insert controls for parity and if men are sexually active.

I appreciate the encouraging remarks of the reviewer. Data on “parity” and “if men are sexually active” could not be fetched from the dataset. 

• I also think it would be good to leave age as a continuous variable, unless you have reasons to believe those three age groups should remain separate

Age was originally a categorical variable in dataset consisting of 5 years age groups. However it was recoded and divided into three classes for the ease of data interpretation. 

• I understand that calculating a Wealth Index using quintiles would automatically separate the observations into five groups with approximately the same number of respondents. So, these phrase is irrelevant: “Each wealth quintile had an approximately similar number of respondents”.

The said phrase has been removed.

• Regarding the logistic regression, I understood from the text that authors only used in the multivariate analysis the variables that had been found to be significant in the bivariate model. However, by looking at Table 3, all variables were included (indeed, they are all significant at the bivariate model). I would add a phrase explaining that they are all significant. If it is not what you did, please, clarify.

All the variables except “employment status” were significant in bivariate analysis. Therefore, rest of the variables were recruited into multivariable model. The modification is present in line no. 162-164.

Conclusion

• I think this manuscript would benefit from a Review as it has the potential to make a good contribution in its field.

We are thankful to the editorial team and worthy reviewers for their valuable time and considering our study for peer review. Your comments are really valuable in improving the manuscript.

Reviewer no. 2:

The question of fertility plays a vital role in many countries’ economic development and health objectives, including Pakistan. It is, therefore, an important area for investigation and the use of large datasets such as the demographic and health survey (PDHS). I appreciate the authors’ curiosity and diligence in investigating this area. Hopefully, my comments can be used to enhance the work.

Introduction

• On page 03, line 51, the reference to fertility reduction as being connected to improvements in health and economic activity is a bit vague. Are you referencing individuals or the state of the country as a whole? Also, it is a bit of a nuanced argument that would require some expansion. 

The said line has been eliminated.

However, your study seems more concerned with reaching the male population and less of the significant picture objectives, perhaps focusing on the male perspective. Why have males historically been left out of the conversation on family planning? Why has it taken so long to recognize the importance of reaching out to males in that space? Why have previous efforts tended to focus on women?

The generalized perception about “contraception being a women’s business” is largely responsible for keeping males from family planning programs target. The importance of male involvement particularly in patriarchal societies was recognized aa few decades back. International conference on population and development held in Cairo was among the initial historical landmark that endorsed the involvement of men in family planning programs. The relevant literature is stated in the Introduction section. 

• On page 4, Line 73 -Line 74, you refer to the United Nations International Conference on Population and Development. That was more than a generation ago, and there have been many iterations of global, regional, and country-specific policies and objectives that have addressed some of the questions you are investigating in this paper. Perhaps you should reference something more recent (e.g., the Millennium Development Goals, followed by the SDGs). Even the UN ICPD had a 25th reunion (i.e., the Nairobi Summit). how have such global structures framed the issue of fertility and contraceptive use behaviors?

The ICPD held in Cairo was an important historical landmark that drew attention towards the role of men in family planning. Furthermore, Reference from international conference on population and development held in Nairobi has also been cited in the introduction (line no. 86-90).

Methods

• The methods read as straightforward and well-executed. I have nothing to add to this section except for the choice of the conceptual framework. Why was that framework chosen? What is the justification for choosing to look at the variables of interest through that lens? Several conceptual frameworks have been used over the years to explain contraceptive use and family planning behaviors (e.g., USAID’s Conceptual Framework and the Women’s and girls’ empowerment in sexual and reproductive health (WGE-SRH) framework). Why did you eschew other frameworks in favor of this one?

USAID conceptual framework is designed to assess the impact of family planning on various areas of women’s lives i.e. personal autonomy, health status, economic resources and acquisition of education etc. Whereas, our study intends to assess the factors that influence contraceptive uptake. Considering the research question and the available data, authors found the employed conceptual framework to be more pertinent.

• As a note, Figure 01 is blurry, and I think you should find a better image that is clearer and easier to read.

The image quality has been improved. Modified image is uploaded with the revised manuscript.

Results

• Be careful of the language you employ in describing your results. For example, in the results section, you use the terms likelihood and odds interchangeably. Though they are used to mean the same thing in colloquial usage, “odds” references a particular relationship between the ratio of probabilities (see pg. 10, Line 171 compared to Line 175).

The issue has been rightly raised. Appropriate changes have been made in the results section (line 203 and 210). 

• Also, be sure to state that a finding is statistically significant and not just significant. Again, statistically significant as a technical term references something particular, while a finding being “significant” can have a more general meaning, such as being generally important or worth noting.

Authors understand the point raised by the worthy reviewer. Findings have been reported appropriately in this regard.

Discussion

The paper’s discussion section is an excellent place to explore your findings and their potential impact. As I read through this section, it mostly reads as a reiteration of the results section. Try exploring some of the following areas:

• Given that in the introduction, you mention that your findings may be used to inform policy, I was hoping the discussion would delve into some commentary about Pakistan’s policy regarding modern contraceptive use if there is one, and how findings can influence that. The fertility challenges have been documented for many decades now and I, as a reader, assume that there are some policies already in place. How do your findings impact those kinds of initiatives? Are men historically excluded from those policies? If men as a subgroup have factored into such initiatives, what do these findings mean?

A brief section has been added in the discussion section that highlights the policy in place for improvement of contraceptive prevalence in Pakistan. The section indicated that there is no clear policy in place that dictates the measures to encourage the involvement of men in family planning and decisions related to contraception (line no. 232-237). 

• On pg. 13, Line 217 to Line 222, you note regional differences, particularly between Punjab and other regions. You theorize that it may be due to improve family planning programs. Is it possible to expand a bit on this? What kind of well-structured family planning programs in Punjab make that place more successful than other regions, and why have such policies not been diffused to other regions.

A brief explanation on regional variation in contraception usage has been added (line no. 261-267)

• Why do you relate your study findings to findings from places like Uganda and Congo? There is nothing wrong per se with that comparison; I don’t think. You could argue that Pakistan occupies a similar socio-economic bracket as those countries, and hence the way they deal with contraceptive use and family planning behaviors may have some bearing on Pakistan. However, neighboring countries in the subregion (e.g., India, Tajikistan, Afghanistan, and Iran) may offer more forthright comparators. I suggest looking through the literature.

The reviewer has rightly pointed out the rationale behind citing literature from places like Tanzania and Congo. Because of similar socioeconomic conditions, the findings from these regions may apply to the subcontinent as well. 

Reviewer no.3: 

Dear Editor,

Thank you for the opportunity to review the manuscript titled Determinants of modern contraceptive use among men in Pakistan: evidence from Pakistan demographic and health survey 2017-18.

Overall, the manuscript is fairly well written and has clear aims. It also focuses on a topic of deep interest to the reproductive health community and is backed by an extensive body of research. However, I find that the justification for the study and its relative contribution to the existing literature is very weak. As such, the unique contribution of these analysis is unclear to me. I think the authors can make a stronger case if they could point out which variables have not been examined, whether is consensus and variables that have inconsistent results. Please see below my specific comments that could further improve the manuscript.

Introduction

Overall – Fairly clear and well written.

• There are multiple typos throughout the manuscript. Authors are suggested to proofread them carefully.

The manuscript has been carefully proof-read. Any typo that was detected has been corrected and highlighted.

• Consider reporting on how Pakistan compares in relation to regional estimates of modern contraceptive use.

The estimates on contraceptive prevalence rate of India and Bangladesh has been added (line no. 56-57).

• Need to define modern contraceptives.

Modern contraceptives, being a dependent variable of the study, has been defined in methodology.

• If effective strategies need to be country-specific, what was the rationale behind estimating prevalence? The rationale part needs to be strengthened.

This study did not estimate any prevalence. With respect I would like to say that the rationale part was written carefully. I would like to refer to the comment of reviewer one that says:

“In the end of the Background section, the authors highlight the importance of their findings to frame family planning programs considering the role of men as decision makers in the matters of family planning and reproductive health in patriarchal societies. I think this is gold and should be brought up again in the results with clear and stated recommendations.”

Methods

• Data sources and sampling techniques – could benefit from better organization structure – information seems randomly placed. Also, expand on stratification and provide a reference for a more elaborate description for the DHS sampling strategy.

The sampling procedure used in PDHS 2017-18 has been added to the methodology section (line no. 109-117). A reference for the details of sampling collection technique used in the survey has been added (line no. 124-125). The information has been placed in a structured format. Information related to PDHS survey, data collection procedure is placed under the section of “data source”. This was followed by the definitions of variables used in the study. Finally. The statistical analysis used in this study was explained.

• 2. Sample –Provide a justification for your sample selection

The sample consisted of 15-49 years old ever married men. This sample was used because the data on contraception was available only for the said bracket of men in PDHS dataset. 

• 3. Please provide another Table and give how each of the independent variables were derived or recoded from the original dataset. This can either be in the manuscript or attached as a supplementary file.

Only two variables (education and age) were recoded. The details on how the said variables were recoded from the original dataset have been incorporated into the manuscript (line no. 151-153)

• 4. Which sampling weight and id were used and was the weight normalized?

Men’s sample weights were used from PDHS dataset. The ID of the weight variable in men’s PDHS dataset was “mv005”. Weights normalization was not carried out.

• What informed the inclusion or selection of the independent variables?

Available data and literature survey guided the recruitment of independent/predictor variables.

• Please specify the model equation

Upon reading various published studies in impact factor journals, it was found that most of the studies do not include the statistical equation in the methodology. As all of the tests are run in software, it is usually not necessary to have an in depth mathematical understanding of the statistical models to run these tests. However, if it is an essential requirement, authors will seek guidance from the statistician to help specify an equation for our model. 

• What informed the choice of the reference categories

Previous studies were explored for this purpose. In addition, the reference categories were selected based on the hypothesis e.g. “residents of Punjab” and “educated men” were anticipated to have a higher modern contraception uptake. Thus, these were selected as reference categories to check how these categories differ relative to their counterparts.

• 8. Did the authors check for multi-collinearity, the results should be provided

Yes. Multicollinearity was checked. The results have been added as a supplementary file named “S1 text”. 

• 9. Please use the STROBE guidelines and present it as an appendix or a supplementary file

STROBE guidelines were used during manuscript write-up. A strobe checklist has been attached as a supplementary file named “S2 text”.

Results

• Consider specifying only the key findings of the study rather than listing all the determinants. Also, use the breakdown of the sentences rather than writing one long sentence that is hard to follow.

Authors respect this concern of the reviewer. The results are already tailored to the key findings. Authors have gone through them in detail. None of them needs to be deleted. In addition, no sentence is long enough to interrupt the continuity of the reader.

• The statistical analysis not been performed rigorously

Critical comments are a great avenue for self reflection and learning. The statistical analysis was reviewed in detail. Reviewer 1 and 2 have also endorsed that data analysis is adequately carried out. Moreover, Authors have tried their level best to respond to the comments of each reviewer raised on statistical analysis. 

Discussion

• Authors started to compare and contrast study findings, which should have been followed after stating the key findings and justification for those findings.

A summary of the study findings is added before discussing each of them separately (line no. 229-232). The justification for all the findings is appropriately provided. Wherever deficient, as indicated by the previous reviewers additional information has been incorporated to give a robust justification of the results. E.g. line 232-237 and line 261-267.

• 2. There is a repetition of most of the findings that are already stated under the results section.

There is no reiteration of the findings per se in the discussion, however these are mentioned in order to compare them with the previous studies of similar kind. 

• 3. Justification for all the discordant results are presented same i.e., due to differences in sample size, study design, setting, and study population. This needs to be study specific rather than a mere generalization.

There is no such recurring justification based on study population, design or sample size in the discussion section. Moreover, various modifications have done in the discussion section which have led to improvements in the justification of reported findings. 

Conclusion

• Clearly and concisely state the conclusions of the study in relation to the key question it sought to answer and the contribution that the paper would make.

The conclusion section has been enriched with the policy recommendation and the importance of its potential to contribute to the improvement of contraception prevalence has been added (line 286-290). 

• 2. The conclusion is well presented. However, the policy implications are not well discussed. The authors can consider beefing them up.

I appreciate the reviewer’s encouraging remark. The policy implication has been added to the conclusion (line 286-290).

---

## [Decision Letter · Decision Letter 1]

30 May 2022

PONE-D-21-23985R1Factors associated with modern contraceptive use among men in Pakistan: evidence from Pakistan demographic and health survey 2017-18PLOS ONE

Dear Dr. Ali,

Thank you for submitting your manuscript to PLOS ONE. After careful consideration, we feel that it has merit but does not fully meet PLOS ONE’s publication criteria as it currently stands. Therefore, we invite you to submit a revised version of the manuscript that addresses the points raised during the review process.

The same 3 reviewers have evaluated the manuscript finding an important improvement but noting some aspects to change in order for the work to be publishable. Note that reviewer 2 comments are in an external file.

We look forward to receiving your revised manuscript.

Kind regards,

José Antonio Ortega, Ph.D.

Academic Editor

PLOS ONE

Journal Requirements:

Reviewers' comments:

Reviewer's Responses to Questions

**Comments to the Author**

1. If the authors have adequately addressed your comments raised in a previous round of review and you feel that this manuscript is now acceptable for publication, you may indicate that here to bypass the “Comments to the Author” section, enter your conflict of interest statement in the “Confidential to Editor” section, and submit your "Accept" recommendation.

Reviewer #1: All comments have been addressed

Reviewer #2: All comments have been addressed

Reviewer #3: All comments have been addressed

2. Is the manuscript technically sound, and do the data support the conclusions?

Reviewer #1: Yes

Reviewer #2: Yes

Reviewer #3: Yes

3. Has the statistical analysis been performed appropriately and rigorously? 

Reviewer #1: Yes

Reviewer #2: Yes

Reviewer #3: Yes

4. Have the authors made all data underlying the findings in their manuscript fully available?

Reviewer #1: Yes

Reviewer #2: Yes

Reviewer #3: Yes

5. Is the manuscript presented in an intelligible fashion and written in standard English?

Reviewer #1: Yes

Reviewer #2: Yes

Reviewer #3: Yes

6. Review Comments to the Author

Reviewer #1: Introduction

I would like to say that I apreciate the authors' effort to improve the paper according to the reviewers' points. It has been completely revised and has improved considerably.

One thing to consider is that by adding the information requested by the reviewers, the manuscript lost a lot of its readability. So, I would spend some time trying to improve the flow of information. For example, line 50 mentions the role of contraceptives in preventing maternal mortality, but line 65 bring the proportions of death. Another example further down: Line 252 to 263 brings several short phrases that can be better connected. See for example, that the word Punjab appears 5 times in this solo paragraph. By the way, you improved this part, but I think we need more descriptions of these regions and their components (because maybe the differences we are observing are only compositional effects).

I would like to strength the point that we have here a paper on a topic that is not very well explored, which is male´s involvement in contraceptive use. Studies inquiring males are absolutely necessary in order to help inform reproductive change policies. Nevertheless, for most modern contraceptives, excluding male condom and vasectomies, women are the ones taking the contraceptives, so, any study of male´s contraceptive behavior is at least partially explained by their female partners´s access and use of contraceptive. This should be mentioned upfront as well as an important consideration which is: women might be taking contraceptives without men knowing about them. So, it is important to either compare this distribution to women´s prevalence or at least mention this statement. You are basically evaluating men´s contraceptive use based on what they know is true.

In my previous review, I listed several points that needed to be observed. In this new submission, the authors addressed one by one, like improving the description of mechanisms of various predictors on contraception, adding background information on each of the regional areas, improving the importance of their findings to frame family planning programs considering the role of men as decision makers and also included a short section explaining why men has been left out of the conversation on family planning (as suggested by the other reviewer). However, I would also add the SDGs as your own justification for this kind of work and I would increase this part in the Discussion section as I will explain below.

As no section “Liteture Review” as presented, the authors brings the literature review at the discussion. I don´t personally like this format, but it is ok. I will comment about it at the “Discussion” section.

Methods

As I mentioned before, it would be important to control for parity and sexual activity. As the authors mention that this information could not be retrieved from the dataset, this limitation should be listed in the discussion. The same goes for age as a continuous variable, that they don´t have. The variables “age” and “education were recoded, but it is necessary to provide the original categories. As for the other ones, it is important to specify that they are being used in their original format.

I also think the analysis should be controlled by marital status. Could part of this ever-married men be widow or separated?

I also would like to see a descriptive table of all contraceptive methods before being aggregate into modern or traditional. Mainly because I would like to see if men who use condoms are different from male who use female´s methods. Aggregating into modern methods may make us lose important variability.

As per another reviewer indicated and it remained missing, I think they should include the information about having used man´s sample weights and the information about choice of reference category in the logistic regressions.

Discussion

Again, I think the incorporation of the reviews made the text lose readability. Check, for example, the first paragraph of the discussion. A great start would be to use “The policy paper…(line 227 to 232) ” and then finish with the lines 223 to 227, which are the empirical findings.

The second paragraph also present structural problems that distract the reader. See how both phrases below can be transformed into a single one:

235 - Discussion with a health worker about family planning enhances the knowledge of contraception that ultimately brings a positive change in behavior [27].

237 - Evidence from behavior change models suggest that knowledge is the 238 first step towards change in behavior [28].

You either flip these phrases or reframe them. This happens throughout the Discussion, so be patient to improve these sentences and paragraphs.

As I pointed in the “Introduction” section, here they keep on adding previous empirical evidence found in the literature with which their findings converse. This is important, as I had mentioned in my first review. But in a journal such as PLOS one, you need to go one step further and discuss why it is important to incorporate men (you cite this on line 231 but do not explore) and how your unique findings help frame family planning programs considering the role of men as decision makers. That means: why is your article unique? How can these findings help Pakistan, which have historically excluded men (as pointed by reviewer 3), create policies that will sucessfully envolve men?

The Conclusion section summarizes all of these really well, but the things above mentioned should be first introduced and dissected in the Discussion sections. They all have mechanisms (i.e. female literacy empower women to demand contraceptives? Or female literacy is associated with male´s literacy, so men who with higher levels of education would be more aware about contraceptive methods and will tend to report more).

There is so much to say here and I think the authors could point directly at how these findings inform policy.

By the way, always keep in mind that the statistics pertain to “men´s perception of contraceptive use”. It is possible that women use contraceptives, yet their husbands don´t know or don´t care to know. So what you are measuring is perception, not use. In my understanding, the only “use” you can measure is vasectomies and condoms. Any data on couple´s discordance on contraceptive methods that you could bring? If not, them set this for future studies.

Reviewer #2: (No Response)

Reviewer #3: The manuscript looks good. The authors addressed all the issues I have raised. I have no further comments on the manuscript

7. PLOS authors have the option to publish the peer review history of their article (what does this mean?). If published, this will include your full peer review and any attached files.

Reviewer #1: No

Reviewer #2: No

Reviewer #3: **Yes: **Dr. Nitai Roy

---

## [Author Response · Author response to Decision Letter 1]

30 Jul 2022

Reviewer #1: 

Introduction

I would like to say that I appreciate the authors' effort to improve the paper according to the reviewers' points. It has been completely revised and has improved considerably.

One thing to consider is that by adding the information requested by the reviewers, the manuscript lost a lot of its readability. So, I would spend some time trying to improve the flow of information. For example, line 50 mentions the role of contraceptives in preventing maternal mortality, but line 65 bring the proportions of death. Another example further down: Line 252 to 263 brings several short phrases that can be better connected. See for example, that the word Punjab appears 5 times in this solo paragraph. By the way, you improved this part, but I think we need more descriptions of these regions and their components (because maybe the differences we are observing are only compositional effects)

I agree with the reviewer’s suggestions. The incorporation of new information in the revised version impaired the continuity of information at certain areas. The said rearrangements have been made (line no. 52-54).Furthermore, referring to the 2nd part of the comment, Line 280-287 have been edited and rephrased.

I would like to strength the point that we have here a paper on a topic that is not very well explored, which is male´s involvement in contraceptive use. Studies inquiring males are absolutely necessary in order to help inform reproductive change policies. Nevertheless, for most modern contraceptives, excluding male condom and vasectomies, women are the ones taking the contraceptives, so, any study of male´s contraceptive behavior is at least partially explained by their female partners´s access and use of contraceptive. This should be mentioned upfront as well as an important consideration which is: women might be taking contraceptives without men knowing about them. So, it is important to either compare this distribution to women´s prevalence or at least mention this statement. You are basically evaluating men´s contraceptive use based on what they know is true.

A similar comment was also raised below. This is a genuine point that I fully agree with. The relevant statements have been written (line no. 303-307).

In my previous review, I listed several points that needed to be observed. In this new submission, the authors addressed one by one, like improving the description of mechanisms of various predictors on contraception, adding background information on each of the regional areas, improving the importance of their findings to frame family planning programs considering the role of men as decision makers and also included a short section explaining why men has been left out of the conversation on family planning (as suggested by the other reviewer). However, I would also add the SDGs as your own justification for this kind of work and I would increase this part in the Discussion section as I will explain below.

I appreciate the encouraging remarks quoted by the worthy reviewer. The comments raised below have been carefully addressed.

As no section “Liteture Review” as presented, the authors brings the literature review at the discussion. I don´t personally like this format, but it is ok. I will comment about it at the “Discussion” section.

The comments in the discussion section have been carefully read and acted upon wherever appropriate. 

Methods

As I mentioned before, it would be important to control for parity and sexual activity. As the authors mention that this information could not be retrieved from the dataset, this limitation should be listed in the discussion. The same goes for age as a continuous variable, that they don´t have. The variables “age” and “education were recoded, but it is necessary to provide the original categories. As for the other ones, it is important to specify that they are being used in their original format.

I agree with the reviewer’s opinion. The raised points have been added as a limitation in the discussion section (line no. 299-307). The readers can refer to the datasets for original categories as the data is freely available on DHS website. 

I also think the analysis should be controlled by marital status. Could part of this ever-married men be widow or separated?

Out of 3691 ever married women, only 63 men were separated, widowed or no longer living together. The data on contraception for these men was missing in the dataset. That’s why we couldn’t control our analysis for marital status (line no. 302).

I also would like to see a descriptive table of all contraceptive methods before being aggregate into modern or traditional. Mainly because I would like to see if men who use condoms are different from male who use female´s methods. Aggregating into modern methods may make us lose important variability.

A descriptive table has been added as a supplementary file that segregates men based on the contraceptives they are using (S3 text).

As per another reviewer indicated and it remained missing, I think they should include the information about having used man´s sample weights and the information about choice of reference category in the logistic regressions.

The information about having used men sample weights has been added (line no. 158-159). Choice of reference category has been explained at line no. 170-174.

Discussion

Again, I think the incorporation of the reviews made the text lose readability. Check, for example, the first paragraph of the discussion. A great start would be to use “The policy paper…(line 227 to 232) ” and then finish with the lines 223 to 227, which are the empirical findings.

The sentences have been rearranged (line no.234-238 repositioned to line no. 249-253).

The second paragraph also present structural problems that distract the reader. See how both phrases below can be transformed into a single one:

235 - Discussion with a health worker about family planning enhances the knowledge of contraception that ultimately brings a positive change in behavior [27].

237 - Evidence from behavior change models suggest that knowledge is the 238 first step towards change in behavior [28].

You either flip these phrases or reframe them. This happens throughout the Discussion, so be patient to improve these sentences and paragraphs.

The phrases have been repositioned (from line no. 259-260 to line no. 256-257). 

As I pointed in the “Introduction” section, here they keep on adding previous empirical evidence found in the literature with which their findings converse. This is important, as I had mentioned in my first review. But in a journal such as PLOS one, you need to go one step further and discuss why it is important to incorporate men (you cite this on line 231 but do not explore) and how your unique findings help frame family planning programs considering the role of men as decision makers. That means: why is your article unique? How can these findings help Pakistan, which have historically excluded men (as pointed by reviewer 3), create policies that will sucessfully involve men?

Further explanation on how the role of men in family planning has evolved over time and why it is important to incorporate them in family planning programs has been added (line no. 243-248) 

The Conclusion section summarizes all of these really well, but the things above mentioned should be first introduced and dissected in the Discussion sections. They all have mechanisms (i.e. female literacy empower women to demand contraceptives? Or female literacy is associated with male´s literacy, so men who with higher levels of education would be more aware about contraceptive methods and will tend to report more).

Authors agree with the reviewers remarks. A plausible explanation/mechanism of each finding has been added at respective places.

By the way, always keep in mind that the statistics pertain to “men´s perception of contraceptive use”. It is possible that women use contraceptives, yet their husbands don´t know or don´t care to know. So what you are measuring is perception, not use. In my understanding, the only “use” you can measure is vasectomies and condoms. Any data on couple´s discordance on contraceptive methods that you could bring? If not, them set this for future studies.

I am thankful to the reviewer for highlighting a very important aspect of this analysis. The statistics of this analysis reflect the predictors men’s behavior/perception towards contraception. If there is any difference between the men who use “male methods” and those who rely on “female methods” could be a potential area for a future study. All the authors appreciate the reviewer for keenly reviewing the manuscript and making a heartful effort to improve the manuscript. The relevant statements have been written (line no. 303-307).

Reviewer’ 2

Overall, great job in addressing my concerns from the first submission. I think the paper is ready to move forward. I would suggest a few more minor edits and things to consider. Please see my comments below.

You did not state the significance level in the Statistical Analysis section. Your audience needs to know the cutoff for statistical significance. I assume it is 0.05?

Level of significance has been stated (line no. 168-169)

Results Consider using two decimal places for the reported statistics even in your tables. Honestly, there is no need to use more than 2 in a study of this type. Furthermore, two decimal places improves readability. I noticed this in Table 3, but consider creating a list of abbreviations/acronyms at the end of your article rather than using superscripts to designate the acronym. For example, FATA means, Federally Administered Tribal Areas. A list of abbreviations/acronyms would make for easy referencing.

The said changes have been made. Abbreviations have been written on line no. 321-326.

Discussion Line 247 - 248 You cannot, or perhaps should not, draw this type of conclusion. Yes, contraceptive use is strongly associated with the man's household socioeconomic bracket (measured by the Wealth Index). Will simply increasing household wealth improve contraceptive use? And who is to improve household income? Since you are looking at population-level data, I assume that will be the government? Are there no other confounding factors that mediate the relationship between wealth and contraceptive 2 use that may offer a better point for policy to address? Lastly, the Wealth Index used in the DHS program is a generalized measure that allows broad comparisons at the population level. Measures of wealth using income and expenditure would give more robust evidence for that association and thus would be better support for that conclusion, which could be a study for another time.

Author’s agree with the reviewer’s stance. The concerned line has been deleted (line no.264-265). 

Your discussion section is missing an element. I think you are missing a paragraph on study limitations.

A detailed paragraph o n limitations has been added in the discussion (line no. 299-307).

 Conclusion Line 286-287 What kind of policy do the authors recommend? Do you have any suggestions s from the references that will help Pakistan achieve a contraceptive prevalence rate of 60% by 2030? I bring this up because you are the experts; you have studied the statistics and provided us with some answers, so from your analysis, what would be your policy recommendations? The conclusion offers you the opportunity to be a bit more specific and direct (this is just a suggestion, as it is fine as written).

Various policy recommendations are mentioned in the conclusion section. A slight modification has been done in line no. 316-317.

---

## [Decision Letter · Decision Letter 2]

18 Aug 2022

Factors associated with modern contraceptive use among men in Pakistan: evidence from Pakistan demographic and health survey 2017-18

PONE-D-21-23985R2

Dear Dr. Ali,

We’re pleased to inform you that your manuscript has been judged scientifically suitable for publication and will be formally accepted for publication once it meets all outstanding technical requirements.

Kind regards,

**Syed Khurram Azmat**, PhD, MPH, MD

*Academic Editor*

PLOS ONE

Additional Editor Comments (optional):

Reviewers' comments:

Reviewer's Responses to Questions

**Comments to the Author**

1. If the authors have adequately addressed your comments raised in a previous round of review and you feel that this manuscript is now acceptable for publication, you may indicate that here to bypass the “Comments to the Author” section, enter your conflict of interest statement in the “Confidential to Editor” section, and submit your "Accept" recommendation.

Reviewer #1: (No Response)

Reviewer #3: All comments have been addressed

2. Is the manuscript technically sound, and do the data support the conclusions?

Reviewer #1: Yes

Reviewer #3: Yes

3. Has the statistical analysis been performed appropriately and rigorously? 

Reviewer #1: Yes

Reviewer #3: Yes

4. Have the authors made all data underlying the findings in their manuscript fully available?

Reviewer #1: Yes

Reviewer #3: Yes

5. Is the manuscript presented in an intelligible fashion and written in standard English?

Reviewer #1: Yes

Reviewer #3: Yes

6. Review Comments to the Author

Reviewer #1: I thank the authors for significantly improving the manuscript. I do not have anything else to add to my comments and if the authors have done what they can to address my previous suggestions, I believe it is good for publication.

My only comment is that I am still missing a brief discussion about how you are actually analysing perceptions when it comes to female methods, especially since you find that men who believe contraception is a women's business are less likely to report using contraception. This seems obvious because they might not know about their women's behavior. So, they are less likely to report, but not necessarily less likely to use (a female method, for example, that he is not aware). As you are not pairing women and men's data, there is no way for you to check whether your finding is true. So, I would rephrase your findings to reflect perception, not real behavior.

Reviewer #3: Author's made vast changes in their manuscript. The manuscript is now quite good for publication in PLOS ONE. No additional comments from my side.

7. PLOS authors have the option to publish the peer review history of their article (what does this mean?). If published, this will include your full peer review and any attached files.

Reviewer #1: No

Reviewer #3: No

---

## [Editor Report · Acceptance letter]

23 Aug 2022

PONE-D-21-23985R2 

Factors associated with modern contraceptive use among men in Pakistan: evidence from Pakistan demographic and health survey 2017-18 

Dear Dr. Ali:

I'm pleased to inform you that your manuscript has been deemed suitable for publication in PLOS ONE. Congratulations! Your manuscript is now with our production department. 

Kind regards, 

on behalf of

Dr. Syed Khurram Azmat 

Academic Editor

PLOS ONE